# Phase-transition tailored nanoporous zinc metal electrodes for rechargeable alkaline zinc-nickel oxide hydroxide and zinc-air batteries

Liangyu Li[1,2], Yung Chak Anson Tsang[3], Diwen Xiao[1], Guoyin Zhu[4], Chunyi Zhi [5] & Qing Chen [1,2,6✉]

Secondary alkaline Zn batteries are cost-effective, safe, and energy-dense devices, but they are limited in rechargeability. Their short cycle life is caused by the transition between metallic Zn and ZnO, whose differences in electronic conductivity, chemical reactivity, and morphology undermine uniform electrochemical reactions and electrode structural stability. To circumvent these issues, here we propose an electrode design with bi-continuous metallic zinc nanoporous structures capable of stabilizing the electrochemical transition between metallic Zn and ZnO. In particular, via in situ optical microscopy and electrochemical impedance measurements, we demonstrate the kinetics-controlled structural evolution of Zn and ZnO. We also tested the electrochemical energy storage performance of the nanoporous zinc electrodes in alkaline zinc-nickel oxide hydroxide (NiOOH) and zinc-air (using Pt/C/IrO$_2$-based air-electrodes) coin cell configurations. The Zn||NiOOH cell delivers an areal capacity of 30 mAh/cm$^2$ at 60% depth of discharging for 160 cycles, and the Zn||Pt/C/IrO$_2$ air cell demonstrates 80-hour stable operation in lean electrolyte condition.

[1] Department of Mechanical and Aerospace Engineering, the Hong Kong University of Science and Technology, Clear Water Bay, Kowloon, Hong Kong, China. [2] The Energy Institute, the Hong Kong University of Science and Technology, Clear Water Bay, Kowloon, Hong Kong, China. [3] George W. Woodruff School of Mechanical Engineering, Georgia Institute of Technology, Atlanta, GA, USA. [4] School of Chemistry and Materials Science, Institute of Advanced Materials and Flexible Electronics (IAMFE), Nanjing University of Information Science and Technology, Nanjing, China. [5] Department of Materials Science and Engineering, City University of Hong Kong, Kowloon, Hong Kong, China. [6] Department of Chemistry, the Hong Kong University of Science and Technology, Clear Water Bay, Kowloon, Hong Kong, China. ✉email: chenqing@ust.hk

Zinc metal, the first-ever battery anode in Alexandra Volta's pile, never ceases to attract research scientists' attention to its unfulfilled potential in a rechargeable battery[1–4]. Being one of the most abundant metals on earth, Zn releases two electrons upon oxidation and offers a theoretical capacity of 3694 Ah/L. The sluggish hydrogen evolution on Zn allows it to work in an aqueous electrolyte[5,6], eliminating fire hazard and lowering the cost. It can be dropped into commercial alkaline Ni cells to replace the expensive metal hydrides. It can also be paired with an air cathode to afford practical specific energy as high as 400 Wh/kg[7–12]. However, neither battery lasts for long under a condition necessary to compete against lithium-ion batteries, which are devices operating with a high areal capacity, a high depth of discharge (DoD), using a limited volume of electrolyte solution[13,14].

The failures root in the undesirable paths of the phase transitions between Zn and ZnO during the electrochemical reactions[15–18]. The most common Zn anode in aqueous alkaline Zn‖NiOOH batteries (historically called Ni-Zn batteries) starts with a bed of mostly ZnO particles, which are partly reduced to Zn metal in the very first charging step. The reduction can invoke long-range transports of zincate ions in the electrolyte to form massive zinc granules that grow cycle by cycle at the cost of the structural uniformity[19]. Upon discharging, the granules will be covered by thick, passivating layers of ZnO similar to anodes that start with large Zn particles[14,16]. Zn mass, in the form of dissolved zincate ions, is drawn to these granules from the rest of the electrode, leaving behind depleted areas, a phenomenon of shape change that has plagued the rechargeability of Zn batteries for decades[17]. We believe that a solution to the issues must achieve two goals together, a uniform formation of Zn metal and a sustainable transition between Zn and ZnO[14,20–24].

Previous research work[8,19,23,25] reveals that a more stable phase transition may take place in bi-continuous (i.e., the solid phase and the void are respectively continuous) porous Zn anodes. The continuous Zn phase, in a morphology of uniformly connected particles, is passivated by a layer of irregular ZnO particles upon deep discharging, which preserve the electronic conductive Zn network at the expense of a minor loss in the accessible capacity. This transition between bare and ZnO-covered porous Zn is stable under practical cycling conditions for more than a hundred cycles[8,25]. However, the laborious fabrication, either through steps of calcination and annealing[8,23] or the preparation of a monolithic salt precursor[25], has limited the scope of the applications. Although it has been reported the spontaneous formation of similar porous Zn in the charging steps of a common ZnO electrode[19], it is not uniform enough to prevent premature cell failure.

Here, we demonstrate an improved electrochemical transition between ZnO and Zn that enables stable cycling of both alkaline Zn‖NiOOH and Zn-air (using Pt/C/IrO2-based electrodes) coin cells. The transition begins by reducing compacted ZnO particles via kinetics-controlled self-organization, from which a network of hundred nanometer-wide Zn ligaments evolves to form a nanoporous (NP) Zn structure. The stability of the NP Zn against electrochemical oxidation is investigated via optical microscopy, electrochemical impedance spectroscopy measurements, and image-based modeling. The NP Zn-based negative electrode enables 300 stable cycles at 40% DoD (20 mAh/cm$^2$) and 160 cycles at 60% DoD (30 mAh/cm$^2$) in alkaline Zn‖NiOOH coin cell configuration and 80 h of stable operation when coupled with an air cathode containing a Pt/C/IrO2 electrocatalyst.

**Fabrication and physicochemical characterization of the nanoporous zinc metal electrode**. The ZnO → Zn transition,

highlighted by a green arrow in Fig. 1a, takes place in an electrolysis cell (Fig. 1b). Commercial ZnO particles are compacted onto a sheet of Sn-plated Cu foam (Supplementary Fig. 1) and then reduced at a fixed voltage towards pure Zn in 3 M KOH electrolyte. The transition from ZnO to Zn proceeds through two steps, $ZnO + H_2O + 2OH^- \rightarrow Zn(OH)_4^{2-}$, and then $Zn(OH)_4^{2-} + 2e^- \rightarrow Zn + 4OH^-$, where $Zn(OH)_4^{2-}$ is the water-soluble zincate ion. To suppress the transport of zincate in the electrolyte that is responsible for non-uniform structural and morphological evolution, the kinetics of the second step must be much faster than that of the first step to maintain a low zincate concentration at the electrode|electrolyte interface. We thus choose 3 M KOH as the electrolyte to slow down the kinetics of the first step and a low voltage (−1.6 V vs. saturated Ag/AgCl) to accelerate that of the second. The reduction starts near the Cu substrate, given the low electronic conductivity of ZnO, and then stays close to the triple-phase boundary of the newly formed Zn, the unreacted ZnO precursor, and the electrolyte. Such a boundary is sustained by the percolation of the metallic Zn phase and the percolation of the electrolyte in the continuous pores, very similar to the electrochemical selective dissolutions of alloys (i.e., dealloying) or metal compounds (i.e., reduction-induced decomposition), generally termed as percolation dissolution[26,27]. The process is distinct from the first charging step of a commonly employed anode in a Zn‖NiOOH battery in the following aspects. ZnO particles in the commonly employed anode are not fully reduced to Zn because of the capacity-limiting cathode. The electrolyte in the battery, if aimed at high DoD, has a KOH concentration much higher than 3 M[28]. The anode often comprises electronic conductive additives and polymer binder for stable, efficient cycling[4,14]. All these differences lead to inhomogeneous reaction in the first charging step of the ZnO anode (illustrated in the left of Fig. 1a), which affects the cell performance stability[14,19].

The above mechanism is supported by both electrochemical and microscopic characterizations. Figure 1c shows a typical chronoamperometric curve for the transition. The current density decays slowly with time until a complete transition of ZnO to Zn, marked by a sudden drop of current to a much lower value contributed by hydrogen evolution. The charge of the reduction in Supplementary Fig. 2 shows that more than 95% of the ZnO particles have been converted into metallic Zn. Instead of being inversely proportional to the square root of time expected for diffusion-limited electrodeposition, the slow decay of the current is likely from the gradual decrease in the area of the reaction front as it propagates from the Cu foam to the top surface, while the kinetics of the self-organization remains stable. Optical microscopy confirms the propagation with the color changes, showing that the initially ivory sample turned black first at the edge, where the black color appears similar to previously reported nanostructures of Zn (Fig. 1d)[25,29].

A difference between the transition and percolation dissolution is the volume expansion of the sample, attributed to the loose connection among the ZnO particles in the compact. The expansion in turn offers a means to control the porosity of the reaction product (Fig. 1b). We place a perforated acrylic plate over the sample, which determines the thickness of the final Zn product assuming that the expansion stops at the plate. The porosity $\varepsilon$ can be estimated with

$$\varepsilon = \alpha \varepsilon_0 \frac{V_{m,ZnO}}{V_{m,Zn}} \qquad (1)$$

where $\alpha$ is the ratio between the foam-to-plate distance and the thickness of the initial ZnO compact, $\varepsilon_0$ is the initial porosity of the ZnO compact, and $V_m$ is the molar volume of the respective

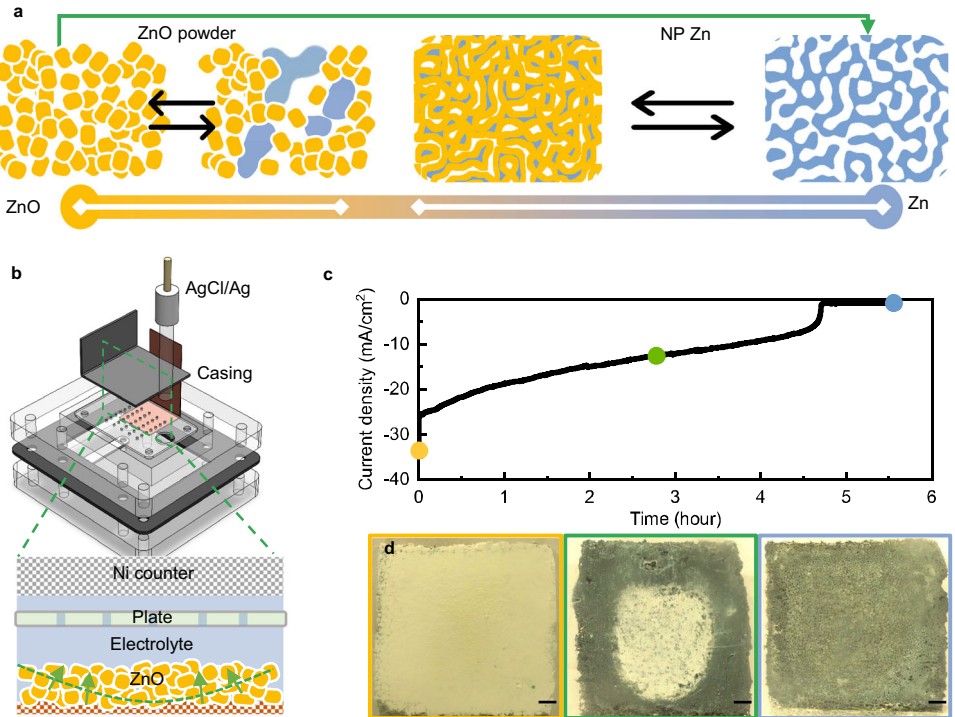

**Fig. 1 Electrochemical investigation on the transition between Zn and ZnO. a** An illustration of the different paths of transition with pure ZnO (yellow) on the left and pure Zn (blue) on the right. In a ZnO powder anode, the transition (black arrows on the left) is between the powder and clusters of Zn metal, whose non-uniform distribution limits the utilization of the Zn mass to <30% (a range marked with a white line at the bottom scale). Instead, the improved transition (the green arrow) takes the ZnO powder to NP Zn, which can then be cycled stably at >40% DoD (black arrows on the right), as highlighted similarly by the corresponding white line at the bottom scale. **b** The electrochemical cell designed for the improved transition through an electrochemical reduction, with the assembled electrolyte chamber highlighted at the bottom. The green arrows indicate the direction of the transition and the green dashed line the reaction front. **c** A typical chronoamperometric curve of the reduction. **d** Photos of the sample at the three stages of the reduction as labeled in **c**. The scale bars are 1 mm.

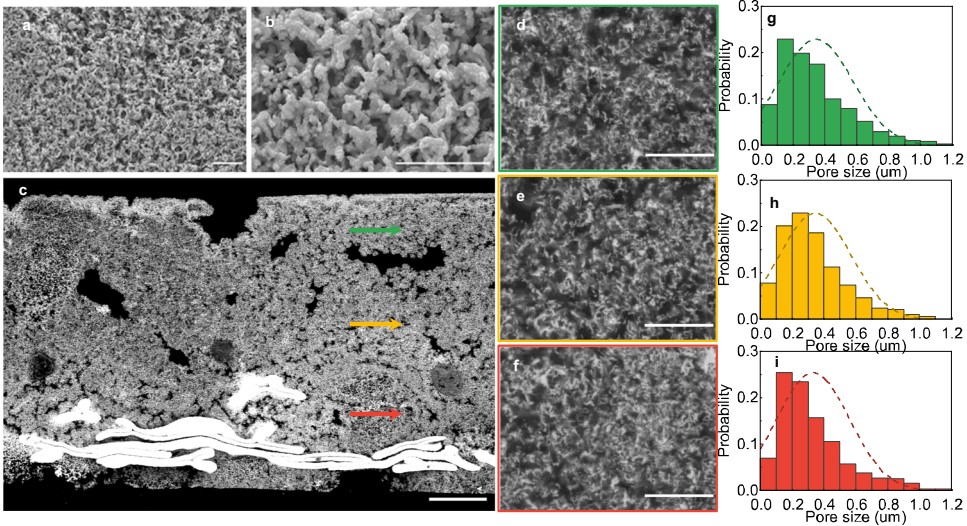

**Fig. 2 Morphological characterization of the nanoporous zinc metal electrode. a, b** SEM images of the surface of the NP Zn under different magnifications. **c–f** Cross-sectional images of the NP Zn filled with the epoxy resin (dark) captured with a back-scattering detector in SEM. The brightest regions at the bottom of **c** correspond to the Sn-coated Cu substrate. Arrows in **c** indicate the positions where **d–f** are captured at a higher magnification, respectively. **g–i** The pore size distributions in **d–f** respectively. The scale bars are 10 μm, except that in **c**, which is 100 μm.

phases. When the plate is placed right at the top surface of the initial ZnO compact ($\alpha = 1$), we will attain the minimum porosity of 36.9%. We instead set $\alpha$ to 2 for a porosity of 73.8%, necessary for the later application as a high-DoD anode.

Figure 2 shows the product of the transition, a bi-continuous NP Zn structure. The surface (Fig. 2a, b), captured by scanning electron microscopy (SEM), comprises connected, porous structure of an average width ~500 nm, as a result of self-organization, most likely

through the diffusion of Zn atoms at or near the solid|electrolyte interface[30]. The composition is confirmed with energy dispersive spectroscopy to be Zn (Supplementary Fig. 3). Pores in the structure are at a similar length scale. We claim the bi-continuity, a characteristic of structures made via percolation dissolution, based on the integrity of the sample and the complete transition to Zn, which require the respective continuities of the solid Zn phase and the pores. The transition from the initial discrete ZnO particles to the porous Zn is underpinned by the ability of the metal surface to accommodate a large population of defects.

To characterize the uniformity in the structure, we filled the pores with epoxy resin and polished the composite for a flat cross-section (Fig. 2c). A back-scattering detector in SEM offers a good contrast between the dark epoxy and the bright metal. There are indeed micrometric pores at the cross-section that were absent in the precursor (Supplementary Fig. 1c). The micrometric pores are unlikely due to the local dissolution of ZnO, which would prefer to occur at the top near the free electrolyte. They are more likely due to the disgregation of the ZnO compact. Zoomed in locally (Fig. 2d–f), the structure observed on the surface prevails throughout the sample. We can further binarize the images (Supplementary Fig. 4) to estimate the size distribution and the area fraction of the pores, the latter of which should be the same as the porosity given the visually isotropic structure. In all the three regions indicated by the green, orange and red arrows in Fig. 2c, the average pore sizes are between 300 and 400 nm (Fig. 2g–i), consistent with the pore size seen on the surface, confirming the uniformity throughout the structure as controlled by the structural organizing kinetics. The estimated porosity falls around 65% (Supplementary Fig. 4), lower than the calculation with Eq. (1) because the areas we selected did not include the micrometric pores.

**Microscopy and spectroscopy characterizations of the nanoporous zinc metal electrode**. The bi-continuous NP Zn structure promises a stable phase transition in an alkaline battery. Unlike a ZnO powder electrode where sporadic clusters of Zn metal can evolve locally or electrodes comprising macrometric metallic Zn powder where thick layers of oxide grow, the NP Zn may retain uniform reactions with its high electronic and ionic conductivities through the two respectively continuous phases. The porous morphology also provides ample sites for the nucleation of ZnO to limit the amount of diffusing zincate that is responsible for morphological changes[30]. We do not expect the structure to return to the ZnO compact in the battery for two reasons. First, in the fabrication, the transition was from pure ZnO to Zn, whereas in the battery, it is between Zn and a mixture of Zn and ZnO (as illustrated in Fig. 1a). Second, the kinetics in the two transitions are different. We design the fabrication to be far from the equilibrium to drive the kinetics of Zn nucleation and surface diffusion that dictates the self-organization of the NP structure. In the battery, the anode under typical conditions (charged at 25 mA/cm$^2$ or lower current) should be much closer to the equilibrium between Zn and ZnO with much lower driving force for the nucleation of new Zn structures and surface diffusion to change the material morphology. If the reaction distribution is uniform, we expect the transition between NP Zn and ZnO during discharging and charging to be the uniform growth and then reduction of ZnO on the Zn surface. However, such transition can be challenged by deep discharging (larger than 40% DoD). The structure may lose the Zn network and thus the electrical conductivity as it converts to ZnO. The growth of ZnO can impede mass transport in the pores; not only does it decrease the porosity, but it also increases the tortuosity of the pores given the irregular morphology of ZnO.

To understand how the structural dynamics may play out in the NP Zn, we apply in situ optical microscopy. It captures the surface changes during discharging and subsequent recharging in a three-electrode cell (Fig. 3a, NP Zn or Zn powder as the working electrode, Zn foil as the counter electrode, Zn wire as the reference electrode, and 6 M KOH with saturated ZnO as the electrolyte). At 20 mA/cm$^2$, the overvoltage is around 0.06 V vs. Zn (blue curves in Fig. 3b). At all stages, the surface displays a uniform color (Fig. 3c) and the morphology conforms to the initial state (the white dashed lines as the approximate, initial boundaries), both in support of a uniform reaction. The structural evolution characterized ex situ under SEM (Fig. 3d and Supplementary Fig. 5), displays similarly the uniformity as we speculated. The Zn structures are covered by a thick layer (~500 nm) of ZnO irregular particles when discharged. Upon recharging, the ligaments are roughened, but the porous network is retained. For comparison, we performed similar in situ characterization of a Zn powder electrode (Fig. 3e), made by pasting commercial Zn powder of an average size ~25 μm on the same substrate (Sn coated Cu foam). The powder electrode could not reach 40% DoD, as the voltage quickly ascended above 0.4 V (yellow curves in Fig. 3b) and invoked the oxidation of the substrate and the electrolyte. The recharging ended in current oscillations at a large overvoltage, consistent with the observation of Zn dendrites and bubbles due to hydrogen evolution (Fig. 3e). The color change is not even. Localized oxidation even exposes the Cu substrate beneath. The shape quickly deviates from the initial state. The differences underline the more uniform reaction distribution in the NP Zn anode owing to the highly connected network.

Electrochemical impedance spectroscopy (EIS) measurements support the claim about the stability of the NP Zn in discharging. At the increment of 5% DoD, we collected spectra in the frequency range of 0.5–200,000 Hz, as we discharged the electrode to 40% DoD in a Swagelok cell with a Ag/Ag$_2$O reference electrode (Fig. 4a). The high-frequency real impedance ($Z_{real}$), contributed partly by the electronic resistance of the electrode, remains low (Fig. 4b). At lower frequency, there is a gradual increase in the impedance, likely contributed by the loss in both the active Zn surface and the effective ionic conductivity in the pores. Nonetheless, the changes are much smaller compared to those of the Zn powder electrode, whose increase in the impedance is drastic even below 10% DoD (Fig. 4c). The EIS curves collected for the full cell (Supplementary Fig. 6) show a similar trend, highlighting the significance of the anode stability.

Underlying the observed stability is the continuous Zn network, which can be separately studied via an image-based simulation (Fig. 4d–g). We binarized a SEM image to attain a representative network of Zn, which then undergo an algorithm of erosion to mimic how it responds to discharging. When 40% eroded, the network of the NP Zn retains its connectivity, visualized by the uniform distribution of local electronic current that can only pass through neighboring Zn pixels under a voltage difference between the top and the bottom of the image. As a comparison, we generated a hypothetical image of randomly distributed, partially overlapped disks to represent an electrode made of Zn particles, instead of using the image of the Zn powder electrode which is different in the length scale and the porosity. The current distribution is not uniform to begin with, given the limited contacts among the Zn particles, and the current fails to pass through at 40% DoD. The connection among the particles breaks down at ~30% DoD in this case, characterized by a rapid drop in the correlation length ($\xi$) of the structure (Supplementary Fig. 7), which is consistent with the prediction by the percolation theory[31]. $\xi$, attained by analyzing the binarized SEM images of NP Zn and the hypothetical images of Zn powder at different

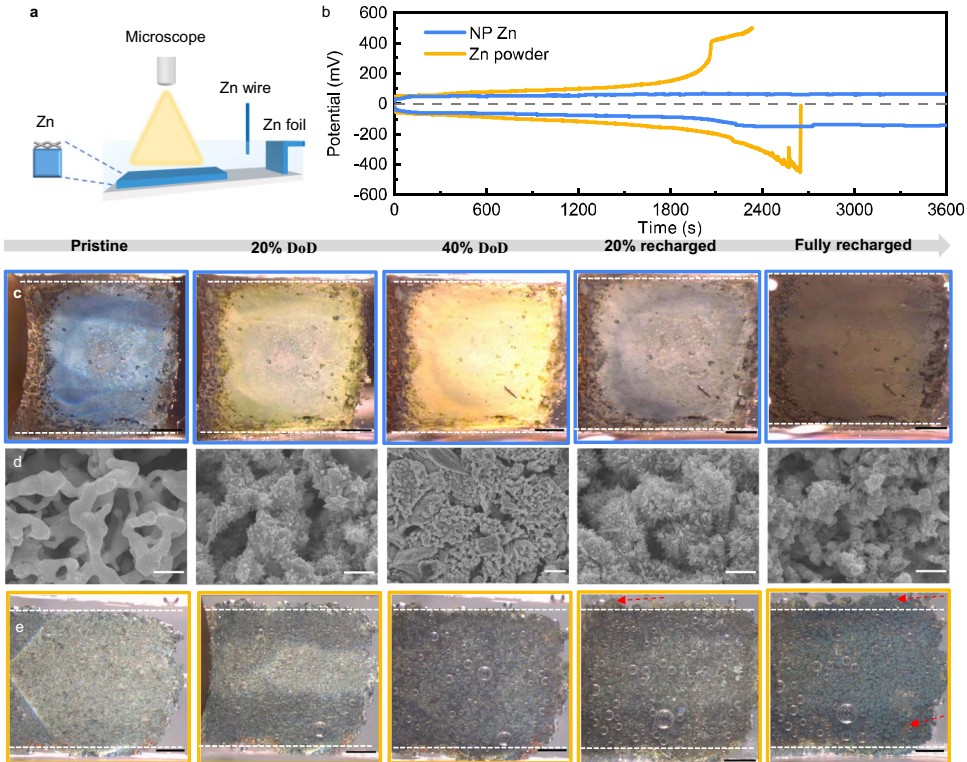

**Fig. 3 Electrochemical and morphological investigations of the two different zinc metal electrodes. a** A schematic of the three-electrode cell for the in situ optical microscopy. **b** Galvanostatic curves of the NP Zn and the Zn powder electrodes during discharging and charging at 20 mA/cm². **c** Photographic pictures of the NP Zn anode captured at five different stages: pristine, 20% discharged, 40% discharged, 20% recharged, and fully recharged, from left to right respectively. The states are estimated based on the theoretical capacities of the electrode. The white dashed lines, whose distance stays the same for all five photographic pictures, correspond to the approximate boundaries of the pristine electrode. **d** SEM of the NP Zn anode harvested from the cell at the same states as the photographic pictures in **c**. **e** Photographic pictures of the Zn powder anode captured at nominally the same states as the NP Zn anode in **c**, where the red arrows highlight locations of morphology changes and the exposure of the Cu substrate. The scale bars are 1 mm in the photos and 1 μm in the SEM images.

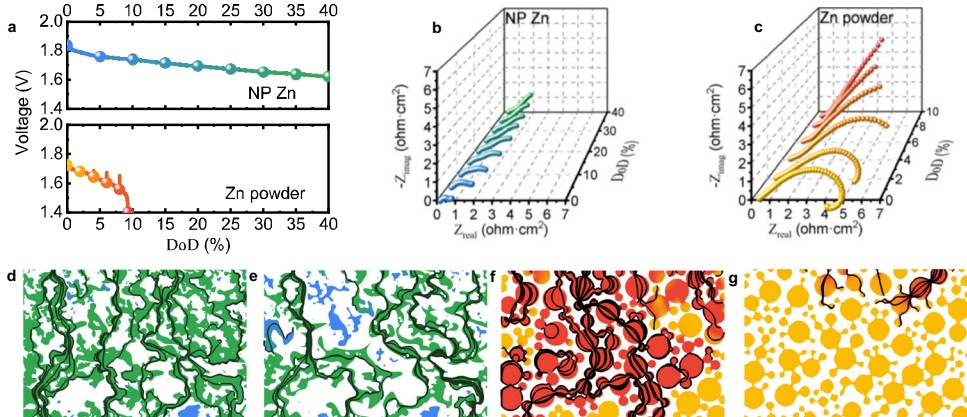

**Fig. 4 Electrochemical characterization and image-based simulation of the two different zinc metal electrodes. a** Discharge curves of NP Zn and Zn powder at 5 mA/cm². At the increments of 5% and 2% for the NP Zn and the Zn powder anodes (shown as solid circles), EIS was recorded and shown in **b** and **c** in the form of Nyquist plots respectively. Image-based simulations of NP Zn with **d** 0 erosion and **e** 40% erosion, and Zn powder with **f** 0 erosion and **g** 40% erosion. The green and red areas in **d**, **e** and **f**, **g** respectively are of high local current when a voltage difference is applied to the top and bottom of the images, whereas the blue and orange areas are of no local current. The current pathways are highlighted in black.

DoD's, corresponds to the average distance between two black pixels (the Zn phase in the binary image) in a cluster of connected black pixels (i.e., a connected structure of Zn). The same drop in $\xi'$ does not happen to the NP Zn until 60% DoD, highlighting its strongly correlated, stable Zn structures.

**Electrochemical energy storage performances of the nanoporous Zn electrode.** A Zn||NiOOH full cell (Fig. 5a) was built with a high loading of Zn (61 mg/cm²) and a low volume of electrolyte (0.6 mL/cm²), both necessary for a high energy density in a practical cell[6,24]. The electrolyte in the cathode and the separator

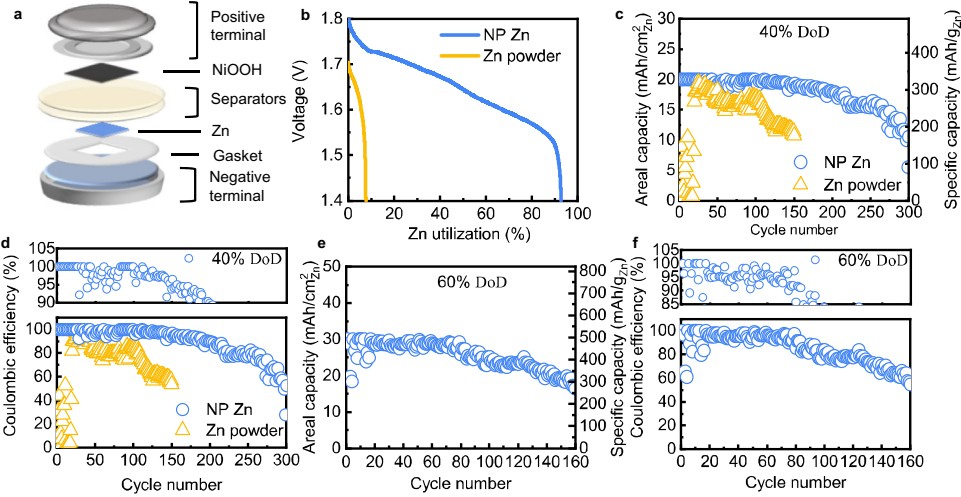

**Fig. 5 Electrochemical performances of Zn||NiOOH full cells. a** Schematic of Zn||NiOOH coin cell used in this study. **b** The Zn utilization of different anodes in primary Zn||NiOOH alkaline batteries. **c** Discharging capacities of the NP Zn and Zn powder anodes at 40% DoD vs. the cycle number. **d** Coulombic efficiency of the NP Zn and Zn powder in the lean electrolyte at 40% DoD, with an additional y-axis (between 90 and 105%) at the top to clearly show the efficiency in the first 100 cycles. **e** Discharging capacities of the NP Zn anode at 60% DoD. **f** Coulombic efficiency of the NP Zn in the lean electrolyte at 60% DoD, with an additional y-axis (between 85 and 105%) at the top to show the efficiency in the first 100 cycles.

contained 6 M KOH and 1 M LiOH, the latter of which prevents anode passivation. The electrolyte in the anode contained additionally 10 wt% $Ca(OH)_2$ to lower zincate solubility[28]. The only impractical component was the cathode, a charged NiOOH-based positive electrode harvested from a commercial battery and oversized so that the cell performance would be dictated by the anode. Another cell, the control group, was assembled with the micrometric commercial Zn powder anode to link the different properties observed in the preceding section directly to the difference in performance. The control group did not contain any ZnO particles in the initial state, as we observed that the addition of ZnO particles to the Zn powder electrode adversely affected the cycling performance (Supplementary Fig. 8). All the specific capacities in the work are reported against the mass of Zn alone, and the current densities are normalized to the area of the Zn electrode. As a primary battery, the cell with the NP Zn anode delivered a specific capacity of 759.32 mAh/g at 5 mA/cm² down to 1.4 V (Fig. 5b). 92.6% of the Zn mass is utilized, as compared to less than 10% of the Zn powder. In a polarization test (Supplementary Fig. 9a), the NP Zn can deliver a peak power density of 264 mW/cm² at a current density of 196.4 mA/cm².

The porous Zn anode is stable at high DoD. At 40% DoD and 25 mA/cm², which correspond to 20 mAh/cm², 328 mAh/g and ~0.5 C, the anode retains its capacity for 100 cycles (Fig. 5c) with a coulombic efficiency of ~100% (upper panel in Fig. 5d). The cell cycling is not affected from hydrogen evolution, which is also shown by the more stable open-circuit voltage of the NP Zn anode compared to that of the Zn powder anode (Supplementary Fig. 9b). We speculate that the rapid formation of ZnO on the Zn metal structure hinders the hydrogen evolution. After 100 cycles, along with the decreasing coulombic efficiency (lower panel in Fig. 5d), the discharging capacity starts to fade at an average rate of ~0.15% per cycle until it drops below 80% of the initial capacity at the 250th cycle (Fig. 5c). In contrast, under the same condition, the Zn powder anode shows worse performances as it cannot reach 40% DoD initially. However, after around 25 cycles a specific capacity increase can be noticed for the electrode comprising the micrometric Zn powder. Although outside the scope of the present research work, we speculate that this effect is associated with structural rearrangement of the active material in the Zn metal electrode.

The connection between the stable phase transition in the NP Zn and the cycling performance is supported by ex situ postmortem morphological investigations. The porous networks are visible in anodes harvested at various stages of the cycling test (Supplementary Fig. 10), except the one at the 300th cycle where the structure hides beneath a layer of hydroxide/oxide difficult to remove without dissolving the electrode during the sampling preparation. Upon the first discharging, the pores get filled with ZnO (Supplementary Fig. 10a) like that in Fig. 3d. The average local porosity should decrease given the volume expansion of the solid phase as Zn oxidizes to ZnO, but we believe there to be ample space for mass transport through the pores as we set $\alpha$ to 2 in the fabrication for a high initial porosity. Upon recharging, the electrode returns to the NP morphology, in support of the proposed path of transition. At the 100th charging cycle, the porous network is covered with ZnO spikes, suggesting one mechanism of capacity fade that would soon kick in; a small fraction of the charging current may produce hydrogen instead of reducing ZnO, which gradually decreases the discharging capacity and aggravates passivation. Future improvements to the anode may include additives that suppress hydrogen evolution, which we have not investigated given the current focus on the benefit of the phase transition.

We can further increase the DoD of the NP Zn anode to 60%, under which it retains 80% of the capacity for over 120 cycles (Fig. 5e and f). A necessary change to the Zn||NiOOH cell is a highly concentrated alkaline electrolyte solution (9 M KOH, saturated with ZnO) to prevent passivation before reaching 60% DoD. The cell delivers a stable areal capacity of 30 mAh/cm² for 80 cycles with a coulombic efficiency larger than 95%. The capacity then fades to 15 mAh/cm² at around the 160th cycle, shorter than the case of 40% DoD, a tradeoff between the cycle life and the capacity utilization for Zn anodes, as we summarize in Supplementary Fig. 11 and Supplementary Table 1. In Supplementary Fig. 11, we include cycling performance from the literature[8,14,22,23,25,32–40] attained at a low ratio of the electrolyte volume over the discharging capacity (<0.5 mL/mAh), a metric advocated by recent work for Zn batteries[24]; only at a low ratio does the utilized Zn mass dominates the weight and volume of a cell. The ratios are 0.03 mL/mAh and 0.02 mL/mAh in our cases of 40% and 60% DoD, respectively. While stable Zn

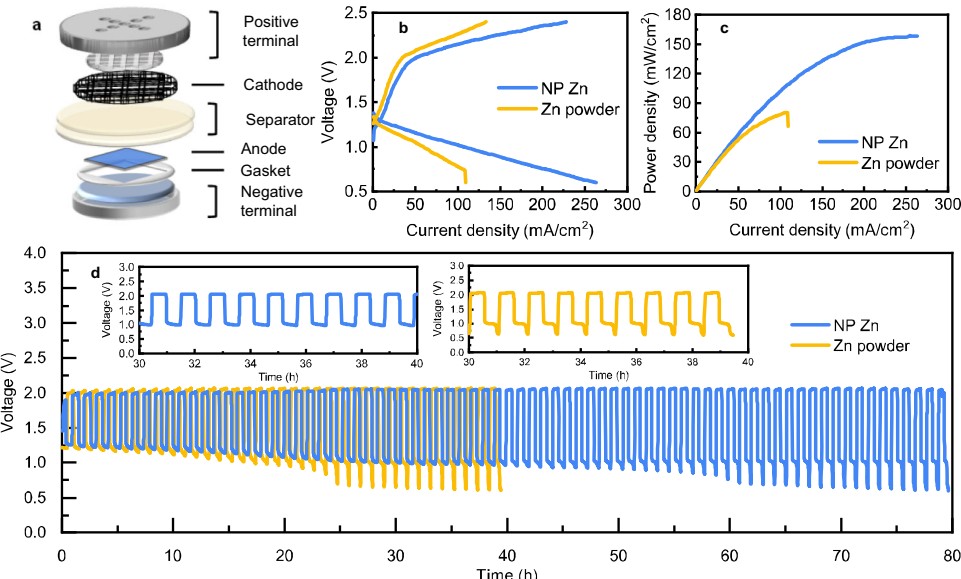

**Fig. 6 Electrochemical performances of the Zn||Pt/C/IrO₂ full cells with NP Zn and Zn powder anodes. a** Schematic of the Zn-air coin cell used in this study. **b** Discharging and charging polarization curves and **c** the corresponding power density curves for discharging. **d** Voltage vs. time curves at 10 mA/cm² and 10% DoD, where the insets highlight those during the 30–40th hours.

batteries have been achieved in the literature under low DoD and low areal capacities, most cells suffered >20% capacity fades before 100 cycles at high DoD.

The NP Zn anode is also tested in Zn-air cell configuration using a Pt/C/IrO₂-based cathode. The cell build (Fig. 6a and Supplementary Fig. 12a) is similar to the Zn||NiOOH cell. It houses a lean electrolyte (9 M KOH) in a coin cell and an air cathode with Pt and IrO₂ as the catalysts. The lean electrolyte is again to ensure a low ratio of the electrolyte volume over the anode capacity (0.03 mL/mAh) so the performance can translate to a practical energy-dense Zn-air battery. The NP Zn anode outputs a current density of 107 mA/cm² at 1.0 V (Fig. 6b) and a peak power density of 159 mW/cm² (Fig. 6c). The overvoltage is significantly smaller than the Zn powder anode. Compared with the Zn||NiOOH cell, a major challenge to the Zn-air cell is the dynamic balance between the limited amount of water in the cell and a humidified oxygen flow at the cathode, which can lead to convective transport of zincate and early passivation. It limits the stable DoD of the NP Zn anode in the cell to 10%, which is 5 mAh/cm², still among the highest values reported for Zn-air cells[41–44]. As shown in Fig. 6d, at 10 mA/cm², the cell discharges at above 1.0 V for more than 50 cycles. Its initial charging and discharging voltages are at 1.97 and 1.26 V respectively with an energy efficiency of 64.47%, and they change to 2.00 and 1.13 V with an energy efficiency of 47.18% after 80 cycles (Supplementary Fig. 12b). The cell accumulates a discharging capacity of 400 mAh/cm² overall for a period of 80 h, before it shows a capacity decay as we cut off the discharging at 0.6 V[45].

In conclusion, we have demonstrated an electrochemical approach to improve the transition between ZnO and Zn for application in alkaline Zn batteries. The research work carried out highlights the importance of the anode microstructure and its dynamics in sustaining stable battery cycling. The characterizations of the NP Zn anode, including microscopy and spectroscopic measurements, offer valuable insights into the root benefits of a material design.

## Methods

**Zinc metal electrodes fabrication.** The bi-continuous NP Zn was fabricated by the electrochemical reduction of ZnO to Zn. Firstly, Sn was deposited on the copper foam (1.6 mm thick, purity >99.99%, MTI Corp) at −1 V (vs Ag/AgCl in saturated KCl) in an electrolyte of 0.1 M SnCl₂ and 0.4 M K₄P₂O₇. The Sn deposited Cu foam was rinsed with deionized water, anhydrous ethanol and dried at 55 °C. Commercial ZnO powder (J&K, >300 nm) was pressed onto the Sn plated Cu foam under 1 ton for 30 s with a hydraulic press. The reduction of ZnO to Zn was performed at −1.55 to 1.6 V (vs Ag/AgCl in saturated KCl) in 3 M KOH with a Ni foam (1.6 mm thick, purity >99.99%, MTI Corp) as counter electrode until the current plateaued. After the reduction, the sample was sequentially placed in acetone (>99.5%) and methanol (>99.9%) for at least 1 h to remove H₂O and KOH and then vacuum dried under 25 ± 5 °C. The as-fabricated NP Zn was directly used as the electrode. For the Zn powder electrode, commercial Zn powder (Aldrich) was mixed with PTFE (5 wt% suspensions) by a mass ratio of 9:1 to form a paste and then pressed onto the Sn deposited Cu foam. For the Zn/ZnO powder electrode, 10% ZnO powder (J&K) was added.

**Physicochemical characterizations.** SEM was conducted with a JEOL-7100F scanning electron microscope equipped with a backscattered electron detector. The dried anode samples were vacuum impregnated with epoxy resin (Struers, Epofix). The epoxy-filled samples were polished with a 0.05 μm aluminum aqueous slurry. The cross-section images with large scale were collected with epoxy-mounted samples coated with a 10 nm layer of gold to prevent charging. Image processing, including the binarization, the pore size analysis was carried out with codes written in MATLAB[46,47]. The dilation and the correlation length were analyzed based on codes in MATLAB's Fractal analysis package[48]. The simulation of current density distribution and current pathways was conducted by applying a voltage difference of 10 mV between the top and bottom boundaries with the electric current interface in the AC/DC module of COMSOL. The in situ optical experiment was performed using a LEICA S9i microscope.

**Electrochemical measurements.** All electrochemical characterizations were carried out with a Biologic VMP3 potentiostat. The polarization test was conducted by linear sweep voltammetry at a scan rate of 1 mV/s. EIS was conducted applying a perturbation of 10 mV under a frequency range from 200 kHz to 0.5 Hz using a three-electrode split test cell (EQ-3ESTC, MTI Corp) which had the same set-up of the coin cell described below (NiOOH as cathode, Zn based electrode as anode), except that an Ag/Ag₂O wire, a quasi-reference electrode, was positioned between the nonwoven cellulose membrane and a Celgard 3501 separator. The Ag/Ag₂O reference electrode was prepared by immersing a Ag wire in a 30% (w/w) hydrogen peroxide water solution for 1 h. Its potential was calibrated to be 0.34 V vs. the standard hydrogen electrode.

Coin cells were used for both the Zn||NiOOH and Zn-air batteries[49]. For the Zn||NiOOH battery, square Zn anodes (0.25 cm²) were assembled into CR2032 coin cells. The NiOOH cathodes (1 cm²) were harvested from fully charged commercial NiMH AAA batteries (~55 mAh/cm², 2600 mAh, GP Batteries) and directly used. In cells reported here, the total cathode: anode capacity ratio was 3:1–4:1. The anode was separated from the cathode by a nonwoven cellulose membrane (16 mm in diameter, 100 μm thick, 75% porosity) and a Celgard 3501 separator (16 mm diameter, thickness: 25 μm, porosity: 55%). For anodes

cycled at 40% DoD, a solution consisting of 6 M KOH + 1 M LiOH was used as the electrolyte for the cathode and separators. The anode was filled with a solution of 6 M KOH + 1 M LiOH with an additional 10 wt% of $Ca(OH)_2$. For anodes cycled at 60% DoD, the aqueous electrolyte was 9 M KOH saturated with ZnO. The total volume of the electrolyte was controlled to be ~150 μL. Cells were galvanostatically cycled with a Neware battery testing station (CT-3008W) at a rate of 25 mA/cm$^2$ (~0.5 C) until they reached a cutoff voltage (1.35 V for discharging and 1.9 V for charging) or a designated capacity (40 or 60% of the theoretical capacity of the anode). If a cell failed to reach the capacity in charging before reaching the cutoff voltage, a step of constant-voltage charging at 1.9 V would be added to reach the capacity or a current below 0.5 mA/cm$^2$. Both the DoD and C-rates were calculated based on the theoretical capacity of the anode. The stability tests of the open-circuit voltage were performed by monitoring the assembled cells at rest (i.e., open circuit voltage).

For the Zn-air batteries, equal weights of Pt/C (20 wt% Pt loading, Sigma-Aldrich) and $IrO_2$ (Sigma-Aldrich) were mixed in a DI water/Nafion/ isopropanol solution, which was then sonicated (120 W) for 60 min. The suspension was dispersed on carbon cloth (CeTech Co., Ltd, W1S1005, 0.785 cm$^2$) via a drop-casting method and dried at 60 °C. The loading of the catalyst is about 5 mg/cm$^2$. The CR2032 coin-type cells were prepared similar to the Zn‖NiOOH battery except the anode size is 1 cm$^2$ with a square shape (50 mAh/cm$^2$). Ni foam (1.6 mm thick, purity >99.99%, MTI Corp) served as the current collector, the spacer, and the spring for the cathode. The electrolyte was 9 M KOH + 5 wt% $Ca(OH)_2$ with a total volume of 150 μL. The galvanostatic cycling was performed inside a sealed gas container (filled with humidified $O_2$) at a current density of 10 mA/cm$^2$, with 30 min for each discharge/charge. The cycling was stopped either when the voltage reached 0.6 V during discharging or 2.5 V during charging. The polarization test was employed under galvanodynamic mode with a scan rate of 0.2 mA/s. No climatic/environmental chamber is used. All the electrochemical measurements are carried out at a temperature of 25 ± 1 °C.

## Data availability

Details on the procedure of image analysis and simulation can be found in the Supplementary Information file. The electrochemical tests' raw data generated in this study have been deposited in Figshare (https://doi.org/10.6084/m9.figshare.19617936, https://figshare.com/s/bd7123526c7b9a0ea3e7). Other information needed is available from the corresponding author upon request.

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

## Acknowledgements

Q.C. acknowledges the funding supports from the Innovation and Technology Fund, Hong Kong (No.: ITS/017/19) and the National Foundation of Natural Science, China (No.: 52022002). L.L., Y.C.A.T. and D.X. acknowledge the technical supports from Materials Characterization and Preparation Facility and Materials Design and Manufacturing Facility at HKUST. Q.C. and C.Z. acknowledge the support from Research Grant Council, Hong Kong (No.: C1002-21G).

## Author contributions

Q.C. supervised the research. L.L. and Y.C.A.T. carried out the fabrication and the simulation. L.L. and D.X. performed the characterizations. L.L. carried out the battery tests. Q.C., G.Z., L.L., and Y.C.A.T. conceived the fabrication strategy. C.Z. advised the tests of the Zn-air batteries. Q.C. and L.L. wrote the paper and all the authors discussed it.

## Competing interests

The authors declare the following competing interests: Q.C., L.L., and Y.C.A.T. filed a U.S. provisional (No.: 63/283,193) and a Chinese (No.: 202111506478.8) patent applications on the fabrication of the nanoporous Zn electrode and its use in Zn batteries.
