## [Peer Review File · Nature Communications]

Reviewers' comments:

Reviewer #1 (Remarks to the Author):

In this manuscript, the authors demonstrate an electrochemical approach to revamp the transition between ZnO and Zn for outstanding performance in alkaline Zn batteries. This work is relatively novel and can attract wide attention. I would recommend the publication of the current work in Nat. Comm. after the following issues addressed.

1. In Figure 1d, the author should explain what is the substance used to turn black first at the edge.
2. In Figure 2c, what is the white part? Please give the description.
3. The authors said that "the estimated porosity falls around 65%" in line 160. The authors should explain how the value was obtained.
4. The authors said that "transition between Zn and ZnO during discharging and charging may thus be the uniform growth and then reduction of ZnO on the Zn ligaments". More detailed explanation or corresponding references should be provided.
5. In line 153, the authors said that "the large holes at the cross-section were due to local disintegration of the ZnO compact bodies". More evidences or corresponding references should be provided.
6. Formatting errors, e. g. lines 176-189.

Reviewer #2 (Remarks to the Author):

In this work, the authors demonstrate a method to revamp the electrochemical transition between ZnO and Zn for a highly rechargeable nanoporous Zn anode. The structure evolves was characterized by in-situ optical microscopy, electrochemical impedance spectroscopy, and image-based simulations. Although this work improved the method of forming the porous material structure of ZnO and Zn mixture, the innovation still could not reach the quality of Nature Communications. It is suggested to try a more professional journal, and the main reasons are listed below:

1. As mentioned by the authors themselves, "The most common Zn anode in a Ni-Zn battery starts with a bed of mostly ZnO particles, which are partly reduced to Zn metal in the very first charging step." in commercial Ni-Zn batteries, the most common zinc anode is the partial reduction of ZnO to a mixture of Zn using a low current densities in the first charging step. The approach used in this article is an improvement on this general practice. There are works have been reported on the use of partially reduced ZnO mixed layers in the anode of alkaline zinc batteries. In recent years, there is also a similar work, such as <https://doi.org/10.1016/j.est.2020.101451>.
2. The layout and structure of this article also need further adjustment. For example, lines 79-99 and paragraphs 176-189 are very untidy.
3. As mentioned in question 1, the current commercial Ni-Zn batteries use a ZnO/Zn hybrid layer. In this paper, there are also some problems in the control group selected by the author. Pure Zn tablets are selected, so obviously there is no comparability in performance.
4. The author by changing the electrochemical reduction ZnO voltage and alkaline solution to control the formation of zinc electrode structure, such as the author described, "The kinetics of the second step, which depends on the applied voltage...We thus choose a large voltage (-1.6 V vs. saturated 91 Ag/AgCl) and 3 M KOH as the electrolyte, whose alkalinity is lower than that of alkaline". As the voltage and solution PH change in the application process, the system will return to the conversion process of ZnO and Zn in commercial Ni-Zn batteries, and the initial state of the electrode cannot be

maintained. In other words, if the initial conditions to maintain the balance of chemical and electrochemical reactions cannot be maintained, the work loses its footing.

5. The author defined (line 179-180) the non-observed color change of the surface in the reaction process as uniform reaction, but there is no evidence to prove the correlation between surface color and uniform reaction, so it is not precise to determine the type of reaction by the absence of color change.

6. The amount of data in this paper is insufficient, some important mechanism characterization has not been carried out, and only optical microscope observation and EIS are used to explain the reaction properties indirectly. Some more scientific methods should be supplemented to prove the uniformity of reaction emphasized in this paper, such as accurate analysis of the micromorphology (SEM\TEM) of the negative electrode at different reaction depths.

Reviewer #3 (Remarks to the Author):

The authors actually synthesized 3D porous Zn-Sn-Cu alloy electrode using a common electrochemical deposition synthesis, although the common structure is claimed bi-continuous nanoporous structure. There lacks powerful evidence on chemical component, electrochemical kinetics to support the structural superiority of Zn electrode and good battery performance. Besides, the battery performance of Ni-Zn and Zn-air batteries is ordinary (A scalable top-down strategy toward practical metrics of Ni-Zn aqueous batteries with total energy densities of 165 W h kg^{-1} and 506 W h L^{-1} , Energy & Environmental Science, 2020,13, 4157-4167). Thus, I oppose it's publication. There are many issues need to be noticed.

1. It is illogical that Zn powder is designed to be the control sample because the alleged Zn anode in fact is alloy electrode. The two electrodes have different chemical component, so the structure merit of Zn-Sn-Cu alloy electrode is difficult to highlight.

2. There are the reported works on Zn-Sn electrode in alkaline battery has clarified their efficient ability of the inhibition of hydrogen evolution and Zn dendrite (Advanced materials interfaces, 2021, 8,2002184; ADVANCED FUNCTIONAL MATERIALS, 2022, 32,2108533; JOURNAL OF ALLOYS AND COMPOUNDS, 2021, 855,157372). The results obtained by In-situ optical microscopy of the surface changes during the discharging and charging are expected.

3. As to battery performances, there lacks electrochemical kinetics data such as hydrogen evolution overpotential, quantitative measurement of hydrogen evolution, GITT curves and diffusivity versus DOD, Linear correlation between the peak current and the square root of the scan rate, V- i correlation and power output of the batteries, etc.

4. Figure S6 appears Zn dendrite after 100 cycles suggests the bad practical performance in the aspects of the stable transition of ZnO to Zn and prolonging the rechargeable life.

The authors actually synthesized 3D porous Zn-Sn-Cu alloy electrode using a common electrochemical deposition synthesis, although the common structure is claimed bi-continuous nanoporous structure. There lacks powerful evidence on chemical component, electrochemical kinetics to support the structural superiority of Zn electrode and good battery performance. Besides, the battery performance of Ni-Zn and Zn-air batteries is ordinary (A scalable top-down strategy toward practical metrics of Ni-Zn aqueous batteries with total energy densities of 165 W h kg^{-1} and 506 W h L^{-1} , *Energy & Environmental Science*, 2020,13, 4157-4167). Thus, I oppose it's publication. There are many issues need to be noticed.

1. It is illogical that Zn powder is designed to be the control sample because the alleged Zn anode in fact is alloy electrode. The two electrodes have different chemical component, so the structure merit of Zn-Sn-Cu alloy electrode is difficult to highlight.
2. There are the reported works on Zn-Sn electrode in alkaline battery has clarified their efficient ability of the inhibition of hydrogen evolution and Zn dendrite (*Advanced materials interfaces*, 2021, 8,2002184; *ADVANCED FUNCTIONAL MATERIALS*, 2022, 32,2108533; *JOURNAL OF ALLOYS AND COMPOUNDS*, 2021, 855,157372). The results obtained by In-situ optical microscopy of the surface changes during the discharging and charging are expected.
3. As to battery performances, there lacks electrochemical kinetics data such as hydrogen evolution overpotential, quantitative measurement of hydrogen evolution, GITT curves and diffusivity versus DOD, Linear correlation between the peak current and the square root of the scan rate, V- i correlation and power output of the batteries, etc.
4. Figure S6 appears Zn dendrite after 100 cycles suggests the bad practical performance in the aspects of the stable transition of ZnO to Zn and prolonging the rechargeable life.

Reviewer #1 (Remarks to the Author):

In this manuscript, the authors demonstrate an electrochemical approach to revamp the transition between ZnO and Zn for outstanding performance in alkaline Zn batteries. This work is relatively novel and can attract wide attention. I would recommend the publication of the current work in Nat. Comm. after the following issues addressed.

We appreciate the recommendation and the insightful comments from the reviewer.

1. In Figure 1d, the author should explain what is the substance used to turn black first at the edge.

We have now added the following sentence to the discussion of Fig. 1d to identify the black substance as NP Zn.

“... where the black color likely stems from nanostructures of Zn (Fig. 1d).”

2. In Figure 2c, what is the white part? Please give the description.

The white, bright part in the image is the Sn-coated Cu foam (more evident in now Fig. S3a). We have now identified it in the figure caption of Fig. 2c.

3. The authors said that “the estimated porosity falls around 65%” in line 160. The authors should explain how the value was obtained.

We thank the reviewer for pointing out this missing information. The porosity was estimated by binarizing the cross-section images and counting the fraction of black pixels, which corresponded to the darker, epoxy-filled pores in the image. We have now included this explanation in Fig. S4.

4. The authors said that “transition between Zn and ZnO during discharging and charging may thus be the uniform growth and then reduction of ZnO on the Zn ligaments”. More detailed explanation or corresponding references should be provided.

The sentence is rather a hypothesis of the path of transition to be confirmed by the later experiments. To avoid confusion, we have revised the sentence as follows.

“If the reaction distribution is uniform, we expect the transition ...”

5. In line 153, the authors said that “the large holes at the cross-section were due to local

disintegration of the ZnO compact bodies”. More evidences or corresponding references should be provided.

Thank you for the suggestion. The conclusion was drawn based on several pieces of evidence. First, there were no large holes in the precursor (Fig. S1c). Second, if the holes were from the dissolution of ZnO into the electrolyte, we would expect them to distribute mainly at the top near the free electrolyte, which was not the case. We are left with the last possibility of disintegration. We have now expanded the sentence as follows.

“There are indeed large holes at the cross-section that were absent in the precursor (Fig. S1c). The holes are unlikely due to the local dissolution of ZnO, which would prefer to occur at the top near the free electrolyte. They are more likely due to the local disintegration of the ZnO compact.”

6. Formatting errors, e. g. lines 176-189.

We appreciate the careful read. We have now fixed the alignment error in the paragraph.

Reviewer #2 (Remarks to the Author):

In this work, the authors demonstrate a method to revamp the electrochemical transition between ZnO and Zn for a highly rechargeable nanoporous Zn anode. The structure evolves was characterized by in-situ optical microscopy, electrochemical impedance spectroscopy, and image-based simulations. Although this work improved the method of forming the porous material structure of ZnO and Zn mixture, the innovation still could not reach the quality of Nature Communications. It is suggested to try a more professional journal, and the main reasons are listed below:

We appreciate the constructive comments from the reviewer. With the responses below, we hope to convince the reviewer that the work provides more than an incremental improvement to the formation of Zn electrodes, but rather a revamped structural transition with fundamental insights and practical significance worthy of publication in Nature Communications.

1. As mentioned by the authors themselves, "The most common Zn anode in a Ni-Zn battery starts with a bed of mostly ZnO particles, which are partly reduced to Zn metal in the very first charging step." in commercial Ni-Zn batteries, the most common zinc anode is the partial reduction of ZnO to a mixture of Zn using a low current densities in the first charging step. The approach used in this article is an improvement on this general practice. There are works have been reported on the use of partially reduced ZnO mixed layers in the anode of alkaline zinc batteries. In recent years, there is also a similar work, such as <https://doi.org/10.1016/j.est.2020.101451>.

We thank the reviewer for bringing this comparison up, which is a very good way for us to clearly explain the significance of our approach. The two do bear similarity. However, there are major differences, as we discuss below, that allow us to call our approach a revamped transition, instead of an incremental improvement.

First, there is a substantial compositional difference. While NP Zn as fabricated has nearly all Zn in its metallic state (Fig. S2), a commercial anode made of mostly ZnO is not and cannot be converted to pure Zn in the charging step in a Ni-Zn battery, unless there is a large excess in the charging capacity of the Ni cathode, which defeats the purpose of high Zn DoD for a high energy density as the Ni cathode has a much lower specific capacity.

Second, assuming that one could convert all ZnO in the commercial anode to Zn, the microstructure would not be the same as NP Zn. Even in its partial conversion in the commercial anode, we have already seen a variety of co-existing Zn structures, including compact layers near a separator and granules in the middle of the electrode (See Ref. # 12 and 17 for examples). The formation of these structures is due to (i) high electrolyte alkalinity necessary for high DoD, which however renders a rapid redistribution of zinc mass via a high concentration of saturated/supersaturated zincate; (ii) additives (e.g., carbon black and polymer binders) necessary for stable, efficient battery performance but also responsible for heterogeneity in the phase transition between ZnO and Zn.

These two differences are critical to the performance, as we have shown the need for a conductive Zn network and a uniform microstructure for the uniform, stable anode reactions. We have now included the above comparison in the first paragraph of the first section as follows.

“The process is distinct from the first charging step of a common anode in a Ni-Zn battery in the following aspects. ZnO particles in the common anode are not fully reduced to Zn because of the capacity-limiting cathode. The electrolyte in the battery, if aimed at high DoD, is much higher in the alkalinity. The anode often comprises conductive additives and polymer binder for stable, efficient cycling. All these differences lead to non-uniformity in the first charging step (illustrated in the left of Fig. 1a), which plants the seed of the later unstable performance.”

2. The layout and structure of this article also need further adjustment. For example, lines 79-99 and paragraphs 176-189 are very untidy.

We acknowledge the shortcomings of the previous writing and have tried to tidy up the two paragraphs.

3. As mentioned in question 1, the current commercial Ni-Zn batteries use a ZnO/Zn hybrid layer. In this paper, there are also some problems in the control group selected by the author. Pure Zn tablets are selected, so obviously there is no comparability in performance.

We chose Zn powder as the control group because (i) it comprises only Zn, the same as the NP Zn anode; (ii) its simplicity allow us to link the structures directly to the performances; and (iii) it delivers a representative performance of Zn anodes at relatively high DoD. We

also note that although commercial batteries use mainly ZnO, an increasing number of studies recently used mainly Zn metal to deliver high DoD (Refs. # 6, 12, 21, and 23 for examples).

Nonetheless, we agree with the reviewer on the need to elaborate on the choice of the control. We have now tested an anode of 85% Zn powder and 10% ZnO particles (Fig. S8). The inclusion of ZnO led to worse performance, the reason behind which can be manifold. For example, ZnO-based anodes are known to perform better in 4 M KOH with KF and K_2CO_3 , whereas Zn-based anodes, particularly those for high DoD, prefer highly alkaline electrolyte (>6 M KOH, which we adopted). The use of low-rate cycles to activate the Zn/ZnO anode still failed to deliver stable performance at 40% DoD, underlying which can be the complex structural evolution discussed in the response to the 1st comment. We have now included this result to highlight the complications introduced by ZnO in the anode and inform the readers the reasons behind our choice of the control group as follows.

“Another cell was assembled with the Zn powder anode to link the different properties observed in the preceding section directly to the difference in performance. We do not have any ZnO particles in the control group, as it complicates the comparison given the potential need for an electrolyte and a cycling protocol different from those for the NP Zn anode (Fig. S8).”

Fig. S8. The cycle performance of Ni-Zn battery with an anode comprising 85% Zn powder and 10% ZnO particles. (a) Specific capacity of the anode vs. the cycle number, attained under the same condition as in Fig. 5b. The anode failed to deliver the designated 40% DoD (red dashed line), likely because the electrolyte (6 M KOH) did not suit ZnO and the poorly conductive ZnO prevented the utilization of Zn. (b) The performance attained by adding 10 cycles at 5 mA/cm² at the beginning, as the common practice to activate ZnO powder anodes.

The performance was better compared to (a), but still inferior to the NP Zn and Zn powder anodes, likely because of the non-uniform structural evolution in the activation process. We thus chose the Zn powder anode as the control group.”

4. The author by changing the electrochemical reduction ZnO voltage and alkaline solution to control the formation of zinc electrode structure, such as the author described, "The kinetics of the second step, which depends on the applied voltage...We thus choose a large voltage (-1.6 V vs. saturated 91 Ag/AgCl) and 3 M KOH as the electrolyte, whose alkalinity is lower than that of alkaline". As the voltage and solution PH change in the application process, the system will return to the conversion process of ZnO and Zn in commercial Ni-Zn batteries, and the initial state of the electrode cannot be maintained. In other words, if the initial conditions to maintain the balance of chemical and electrochemical reactions cannot be maintained, the work loses its footing.

We sincerely thank the reviewer for this comment, which drives us to think deeper about our strategy. In addition to the structural analysis and the cycling performance that support the stability of the structure, we would like to offer a discussion on the fundamental differences between the structural transitions in the fabrication and in the battery operation.

First, the two transitions are between different pairs of states. In the fabrication, we convert pure ZnO to pure Zn, whereas the transition in the battery, as determined by the voltage, is between pure Zn and a mixture of Zn and ZnO. In the battery, the electrode is never brought to the initial state of pure ZnO, which can certainly destroy the NP structure. (It is now also emphasized in the caption of Fig. 1a, a good place to illustrate the different states.)

Second, the kinetics of the two transitions are different. We design the fabrication to be far from the equilibrium to drive the kinetics of Zn nucleation and surface diffusion that dictates the self-organization of the NP structure. In the battery, the anode under the tested condition should be much closer to the equilibrium with little nucleation of new Zn structures and no driving force for surface diffusion to reshape the structure. The only outlier in this hypothesis is the diffusion of soluble zincate in the highly alkaline battery electrolyte. Fortunately, the precipitation of ZnO from zincate seems rapid on the Zn microstructures to minimize the impact of zincate diffusion, consistent with previous observations (Refs #6, 21, and 23).

While the experimental results are still the strongest support of the stability, we have now included the above discussion at the beginning of the second section to lay the ground.

“We do not expect the structure to return to the ZnO compact in the battery for two reasons. First, in the fabrication, the transition was from pure ZnO to Zn, whereas in the battery, it is between Zn and a mixture of Zn and ZnO (as illustrated in Fig. 1a). Second, the kinetics in the two transitions are different. We design the fabrication to be far from the equilibrium to drive the kinetics of Zn nucleation and surface diffusion that dictates the self-organization of the NP structure. In the battery, the anode under typical conditions should be much closer to the equilibrium with little nucleation of new Zn structures and no driving force for surface diffusion to reshape the structure.”

5. The author defined (line 179-180) the non-observed color change of the surface in the reaction process as uniform reaction, but there is no evidence to prove the correlation between surface color and uniform reaction, so it is not precise to determine the type of reaction by the absence of color change.

We acknowledge the inadequacy of the surface color change to support the uniform reaction. We have revised the discussion to instead highlight the dimensional and structural changes. We added more precise evidence. The first is a semi-quantitative analysis of the electrode dimensions during the in-situ characterization (revised Fig. 3c and e). Using the exposed Cu substrate as an internal scale, we highlight the boundaries of the electrode at the pristine state as white dashed lines. The NP Zn anode conformed to the boundaries, whereas the Zn powder anode quickly deviated from them during the in-situ characterization, demonstrating the distinct reaction distributions between the two. The second is the SEM characterization of the NP Zn anode at all the states and the statistics of the ligament size (Fig. 3d and S5), showing the retention of the uniform NP structure. The paragraph of the discussion has now been revised as follows.

“At all stages, the surface displays a uniform color (Fig 3c) and the shape conforms to the initial state (the white dashed lines as the approximate, initial boundaries), both in support of a uniform reaction. The structural evolution at a finer scale, characterized ex-situ under SEM (Fig. 3d and S5), displays similarly the uniformity as we hypothesized. The Zn ligaments are covered by a thick carpet of ZnO spikes when discharged. Upon recharging, the ligaments are roughened, but the porous network is retained... Localized oxidation even exposes the Cu substrate beneath. The shape quickly deviates from the initial state. The differences underline

the more uniform reaction distribution in the NP Zn anode owing to the highly connected network.”

“Fig. 3. In-situ optical microscopy and ex-situ electron microscopy of the changes during the discharging and charging of two Zn anodes. (a) A schematic of the three-electrode cell for the in-situ optical microscopy. (b) Galvanostatic curves of the NP Zn and the Zn powder electrodes during discharging and charging at 20 mA/cm^2 . (c) Photos of the NP Zn anode captured at five different stages: pristine, 20% discharged, 40% discharged, partly recharged, and fully recharged, from left to right respectively. The states are estimated based on the theoretical capacities of the electrode. The white dashed lines, whose distance stays the same for all five photos, correspond to the approximate boundaries of the pristine electrode. (d) SEM of the NP Zn anode harvested from the cell at the same states as the photos in (c). (e) Photos of the Zn powder anode captured at nominally the same states as the NP Zn anode in (c), where the red arrows highlight locations of large shape changes and the exposure of the Cu substrate. The scale bars are 1 mm in the photos and $1 \mu\text{m}$ in the SEM images.”

“Fig. S5. SEM images of the NP Zn anode at the same stages in the corresponding images in Fig. 3d, but at lower magnifications to show the uniformity. Below each image is a statistical analysis of the ligament size based on 25 randomly chosen ligaments from the image. The mean value increases in charging and decreases back in discharging, while the distribution is relatively stable.”

6. The amount of data in this paper is insufficient, some important mechanism characterization has not been carried out, and only optical microscope observation and EIS are used to explain the reaction properties indirectly. Some more scientific methods should be supplemented to prove the uniformity of reaction emphasized in this paper, such as accurate analysis of the micromorphology (SEM\TEM) of the negative electrode at different reaction depths.

We appreciate the suggestion on more characterization. As discussed in the preceding response, we have now characterized the microstructures at different reaction depths with SEM (Fig. 3d and S5), which confirm the hypothesized the path of phase transition and support the uniform reaction. We have also included more quantitative analysis of the structural evolution, including the shape change estimates (Fig. 3c and e) and the ligament size distribution (Fig. S5). We have also included EDS results to show the composition of NP Zn (Fig. S3) and EIS of the battery to further demonstrate the importance of the anode structure (Fig. S6). We do not consider TEM necessary as the length scale and the (non)uniformity of Zn anodes reside both at scales much larger than a few nanometers.

Fig. S6. EIS of the full cell with (a) the NP Zn anode and (b) the Zn powder anode, attained under the same condition as in Fig. 4b and c.”

Reviewer #3 (Remarks to the Author):

The authors actually synthesized 3D porous Zn-Sn-Cu alloy electrode using a common electrochemical deposition synthesis, although the common structure is claimed bi-continuous nanoporous structure. There lacks powerful evidence on chemical component, electrochemical kinetics to support the structural superiority of Zn electrode and good battery performance. Besides, the battery performance of Ni-Zn and Zn-air batteries is ordinary (A scalable top-down strategy toward practical metrics of Ni-Zn aqueous batteries with total energy densities of 165 W h kg^{-1} and 506 W h L^{-1} , Energy & Environmental Science, 2020,13, 4157-4167). Thus, I oppose it's publication. There are many issues need to be noticed.

We need to first refute the reviewer's assertion on the composition. It is evident from the fabrication mechanism that the anode is not an alloy. The active material is Zn, supported by a Sn-coated Cu current collector. No alloy can form, as no atoms can diffuse between the bulk solid phases at room temperature and no Sn or Cu can dissolve into the electrolyte at the low voltage. We have now included EDS of the cross-section and the surface (Fig. S3) to show the composition of NP Zn and the clear separation between Zn and the Cu substrate.

“Fig. S3. Composition analysis of the NP Zn. (a) EDS of the cross-section of the NP Zn filled with the epoxy with the distribution of Zn, Cu and Sn colorized according to the bottom left. (b) EDS of the surface of the NP Zn and (c) the corresponding quantitative compositional analysis. Excluding C, Zn, K, and O were identified as the main components, the latter two of which were likely from the residue of electrolyte difficult to completely remove especially from the pores beneath. Nonetheless, no Cu can be found in the surface structure.”

Second, we need to argue against the phrase ‘common’ that appeared twice in the comment. Whereas a common deposition converts metal ions in an electrolyte to metal, our approach uses a ZnO compact as the source of metal ions for the formation of the NP structure. While we do not understand the meaning of a ‘common’ structure, we claim the bi-continuity for definitive reasons. The pores are continuous, otherwise there would be many residual ZnO particles. The metal ligaments are continuous, otherwise the structure would fall apart. The structure is superior to beds of particles or fibres, which we would consider common, as they require additives or high temperature sintering, which introduce heterogeneity or coarsening that can adversely affect the electrode properties.

Third, we focused the characterizations on factors that matter, including the morphology and the electrical conductivity. They may not appear ‘powerful’, but they address the key structure-property-performance relationships. The chemical composition characterization is now added to clear the confusion (Fig. S3). As for the electrochemical kinetics, we believe that it has already been shown in the polarization (Fig. 6 and S9) and EIS curves (Fig. 4).

At last, we do believe that the performance stands out, as shown by Fig. 5d which is consistent with previous summaries (Ref. #12 and Sustainable Energy Fuels, 2020, 4, 3363). The work mentioned in the comment brings us more confusion than insights. It was focused on Ni cathodes but showed more than 3000 cycles at a high nominal areal capacity in a Ni-Zn coin cell. The anode was merely a layer of Zn deposited on Cu mesh. Cu mesh or foam has been the substrate of numerous alkaline Zn anodes, but none registered such performance. In addition, the cathode was assembled at the discharged state, which means that the anode capacity was from zincate instead of the Zn metal, contradicting the listed electrolyte volume of 100 μ L, which cannot provide the said capacity. Later in the work, the anode was changed to a more typical mixture of Zn, ZnO and additives for pouch cells, although other critical

information, including the electrolyte volume, became missing. We consider the above reasons strong enough to put the Zn anode performance in this report in doubt.

1. It is illogical that Zn powder is designed to be the control sample because the alleged Zn anode in fact is alloy electrode. The two electrodes have different chemical component, so the structure merit of Zn-Sn-Cu alloy electrode is difficult to highlight.

We have refuted the assertion of the composition earlier. To ensure a fair comparison, we in fact used Sn-coated Cu as the current collector of the Zn powder anode, too, not because of any alloy effect but to ensure that the differences in the properties and the performance root in the different microstructures.

2. There are the reported works on Zn-Sn electrode in alkaline battery has clarified their efficient ability of the inhibition of hydrogen evolution and Zn dendrite (*Advanced materials interfaces*, 2021, 8,2002184; *ADVANCED FUNCTIONAL MATERIALS*, 2022, 32,2108533; *JOURNAL OF ALLOYS AND COMPOUNDS*, 2021, 855,157372) . The results obtained by In-situ optical microscopy of the surface changes during the discharging and charging are expected.

Besides the refuted assertion on the composition, the three papers mentioned here have little to do with our work. The first two dealt with neutral Zn batteries. The last applied a Zn-alloy to Li- and Na-ion batteries. The design principles of neutral Zn anodes are very different, as their charging and discharging processes involve no ZnO. Instead, because of the near neutral pH, hydrogen evolution becomes a bigger issue.

3. As to battery performances, there lacks electrochemical kinetics data such as hydrogen evolution overpotential, quantitative measurement of hydrogen evolution, GITT curves and diffusivity versus DOD, Linear correlation between the peak current and the square root of the scan rate, V- i correlation and power output of the batteries, etc.

We do not consider there to be a lack of data to corroborate the performance. The polarization curves (V-i correlation) and the power outputs of the batteries were already in the manuscript (Fig. 6 and S9), which demonstrated the rapid kinetics of the NP Zn anode. Whereas GITT and the correlation between peak currents and scan rates of cyclic voltammetry can characterize rates of diffusion, there is no sign that the rates of diffusion are limiting the performance of our anodes. We do not want to overload the work with unnecessary data to distract readers' attention from the key points.

We do agree with the reviewer on the need for evidence on hydrogen evolution. We have now included data on the stability of the anode at its open circuit voltage (Fig. S9e), a way of evaluating the hydrogen evolution, which can spontaneously oxidize Zn and cause a gradual decrease in the battery voltage. We saw a very stable voltage in the case of NP Zn, likely because the large Zn surface was rapidly covered with ZnO and thus passivated against catalyzing hydrogen evolution, an attribute that also mitigated the shape change.

“Fig. S9. Additional data of the Ni-Zn batteries. (a) The Zn utilization of different anodes in primary Ni-Zn batteries. (b) Polarization curves of different anodes in Ni-Zn batteries. (c) Coulombic efficiency of the NP Zn and Zn powder in the lean electrolyte at 40% DoD. (d) Coulombic efficiency of the NP Zn in the lean electrolyte at 60% DoD. (e) The stability of the

open-circuit voltage of Ni-Zn batteries over time, an indicator of the severity of hydrogen evolution.”

4. Figure S6 appears Zn dendrite after 100 cycles suggests the bad practical performance in the aspects of the stable transition of ZnO to Zn and prolonging the rechargeable life.

The sharp bright spikes in the original Fig. S6 (now Fig. S10c and d) are ZnO instead of Zn. They readily appear in the first discharging cycle, when no Zn dendrite would grow (See Fig. 3d and Ref. #6 and 23 for examples). The morphology itself also reveals its chemical nature. Zn dendrites are larger in the length scale and lower in the aspect ratio because of the highly mobile Zn atoms on the surface and the crystal structure. The dendrites should also comprise multiple levels of fractals distinct from these bundles of spikes (See Ref. #18 for examples).

REVIEWERS' COMMENTS

Reviewer #1 (Remarks to the Author):

Since all the issues concerned have been clarified in the present version of manuscript, I therefore suggest acceptance of this paper for publication in Nature Communications.